# A Narrative Review of the Efficacy of Long COVID Interventions on Brain Fog, Processing Speed, and Other Related Cognitive Outcomes

**DOI:** 10.3390/biomedicines13020421

**Published:** 2025-02-10

**Authors:** Bryana Whitaker-Hardin, Keith M. McGregor, Gitendra Uswatte, Kristine Lokken

**Affiliations:** 1Neuroscience Theme, Graduate Biomedical Sciences Doctoral Training Program, Joint Health Sciences, University of Alabama at Birmingham, Birmingham, AL 35294, USA; whitakeb@uab.edu; 2Birmingham Veterans Affairs Geriatric Research Education and Clinical Center, Birmingham Veterans Affairs Health Care System, Birmingham, AL 35294, USA; kmmcgreg@uab.edu; 3Department of Clinical and Diagnostic Sciences, School of Health Professions, University of Alabama at Birmingham, Birmingham, AL 35294, USA; 4Departments of Psychology & Physical Therapy, College of Arts and Sciences, University of Alabama at Birmingham, Birmingham, AL 35294, USA; guswatte@uab.edu; 5Department of Psychiatry and Behavioral Neurobiology, Heersink School of Medicine, University of Alabama at Birmingham, Birmingham, AL 35294, USA

**Keywords:** long COVID, PASC, cognition, brain fog

## Abstract

In the years following the global emergence of severe acute respiratory syndrome coronavirus 2 (SARS-CoV-2), or COVID-19, researchers have become acutely aware of long-term symptomology associated with this disease, often termed long COVID. Long COVID is associated with pervasive symptoms affecting multiple organ systems. Neurocognitive symptoms are reported by up to 40% of long COVID patients, with resultant effects of loss of daily functioning, employment issues, and enormous economic impact and high healthcare utilization. The literature on effective, safe, and non-invasive interventions for the remediation of the cognitive consequences of long COVID is scarce and poorly described. Of specific interest to this narrative review is the identification of potential interventions for long COVID-associated neurocognitive deficits. Articles were sourced from PubMed, EBSCO, Scopus, and Embase following Preferred Reporting Items for Systematic Reviews and Meta-Analyses (PRISMA) guidelines. Articles published between the dates of January 2020 and 30 June 2024 were included in the search. Twelve studies were included in the narrative review, including a feasibility study, a pilot study, a case series, a case study, and an observational study, in addition to three randomized clinical trials and four interventional studies. Overall, treatment interventions such as cognitive training, non-invasive brain stimulation therapy, exercise rehabilitation, targeted pharmacological intervention, and other related treatment paradigms show promise in reducing long COVID cognitive issues. This narrative review highlights the need for more rigorous experimental designs and future studies are needed to fully evaluate treatment interventions for persistent cognitive deficits associated with long COVID.

## 1. Introduction

Coronavirus disease 2019 (COVID-19) is caused by the severe acute respiratory syndrome coronavirus 2 virus (SARS-CoV-2) and can trigger a diverse range of symptoms that manifest into mild, severe, or even fatal illness [1]. Since the onset of the COVID-19 pandemic in December of 2019, more than 770 million cases of COVID-19 have been reported to the World Health Organization [2]. This unprecedented impact on global healthcare has continued to evolve, with one of the newest developments being the persistence of debilitating symptoms months (or years) after an initial infection of SARS-CoV-2. Also known as long COVID, chronic COVID, long-haul COVID, and/or post-acute sequalae of COVID-19 (PASC), the long-term presence of multisystemic symptoms such as cognitive dysfunction (“brain fog”), dysautonomia, respiratory problems, dyspnea, musculoskeletal pain, and fatigue and/or post-exertional malaise is a severe burden on patients [3,4,5,6,7] (Long COVID is estimated to affect approximately 3.4–6.9% of adults and 1.3% of children living in the United States who have been previously infected with SARS-CoV-2 [8,9] with cumulative global incidence of long COVID as high as 400 million individuals and an estimated annual economic impact of approximately USD 1 trillion, equivalent to about 1% of the global economy [10].

Long COVID symptoms can result in a loss of daily functioning, including interfering with work as well as issues with returning to work, and long-term physical manifestations across multiple organ systems [9,11]. These functional and physical impairments are often accompanied by neurocognitive impairments, a decrease in overall quality of life, and worsening or novel onset of mental health conditions [12].

The term “brain fog” has garnered attention as the shared cognitive experience of a heterogeneous set of cognitive issues, including intermittent confusion, effortful cognition, brain-based fatigue, slowed processing speed, executive function deficits, forgetfulness, and sluggish thinking or clouded mentation by those with long COVID [13]. Long COVID brain fog can be exacerbated by or lead to insomnia, depression, and anxiety disorders [6,14]. Brain fog symptoms are also reported by others with infection-associated chronic conditions (referred to as post-acute infection syndromes) and patients with seemingly unrelated neurological, psychiatric, and physical conditions including mild traumatic brain injury (mTBI), multiple sclerosis (MS), lupus, myalgic encephalomyelitis (ME), bipolar disorder, and celiac disease, among others [15]. There is no scientific definition of brain fog to date; however, it is broadly understood in the literature as generalized cognitive dysfunction, subjectively related to issues with memory, attention, and executive functioning [16].

Although the impact of brain fog on cognition is typically of low to moderate severity, the impact on daily functioning, work capacity, and quality of life can be devastating, especially considering patients reporting these symptoms are generally younger and of workforce age [17]. Additionally, patients with brain fog symptoms can put an extensive burden on the medical system through increased healthcare utilization. The literature and the lay media are riddled with reports of long COVID patients feeling they must “prove” their illness, often leading to lengthy diagnostic odysseys with few effective treatment options [18].

Much like its diverse symptomatic presentation, several pathophysiological mechanisms have been suggested to contribute to the neuropsychological deficits in long COVID. As highlighted in Mehandru and Merad’s work, chronic inflammation, marked by elevation in proinflammatory cytokines, is a hallmark pathology that that is associated with the significant persistence and severity of long COVID-associated neurological symptoms (2022) [19]. Importantly, neuroinflammation markers as assessed with positron emission topography (PET) scans have been found across multiple brain regions in individuals with long COVID [20]. Additional mechanisms that have been proposed to specifically contribute to cognitive deficits in this population include reduced cerebral blood flow as shown by arterial spin labeling magnetic resonance images (MRIs), central nervous system atrophy (e.g., loss of hippocampal neurogenesis), autonomic system dysfunction, and reduced serotonin levels [21,22,23,24,25,26]. Autonomic system dysfunction is often the result of impaired vagus nerve function, but the exact molecular mechanisms are unknown [27]. Other general pathophysiological mechanisms that are suggested to contribute to long COVID-associated neurological symptoms include metabolic dysfunction, viral persistence, and systematic organ damage from acute COVID-19 infection [28,29,30].

The literature on the remediation of neuropsychological consequences of long COVID is scarce and poorly described. Methodological concerns are common, including heterogeneity of patients and ill-described procedures, including variable time frame post-infection and use of low-sensitivity screening measures. To this end, studies examining potential interventions to reduce or alleviate neuropsychological symptoms are also limited and often present with contradictory conclusions. However, patients and providers are in search of non-invasive, safe, and efficacious interventions to improve processing speed and other cognitive issues post-COVID. Certainly, effective therapies to address the sequalae of long COVID, particularly the cognitive symptoms, are a priority for identification and dissemination. Of specific interest to this narrative review is the identification of potential remediation interventions for long COVID-associated neurocognitive deficits, which are present in an estimated 20–40% of long COVID cases [31]. The authors chose to conduct a narrative review, as few papers that focused on the treatment of cognitive issues in long COVID included comparable outcome measures or detailed quantitative outcomes, making it impossible to extract and standardize data from each study to statistically pool the data to generate an estimate of the effect size across the studies.

It seems important to investigate the different avenues of treatment for individuals dealing with long COVID. For the purposes of this narrative review, we examine if long COVID-related cognitive symptoms such as brain fog and processing speed problems were measured and potentially alleviated in psychological/cognitive training, non-invasive brain stimulation therapy, exercise rehabilitation, and pharmacological and/or other related treatment paradigms. We chose to present the material in a narrative review format, as the primary goal of this paper was to provide a comprehensive overview of currently published long COVID interventions to inform a cohesive strategy for future treatment interventions.

## 2. Materials and Methods

We searched three databases: PubMed, Embase, and Scopus. Search terms were as follows: (intervention OR rehabilitation) AND (“PASC” OR long covid OR post-covid OR chronic covid) AND (brain fog or processing speed). Articles published between the dates of January 2020 and 30 June 2024 were included in the search. The aspirational goal was to include only randomized controlled clinical trials in this review; however, given the dearth of literature examining interventions for long COVID cognitive issues, a variety of methodologies were considered for inclusion. After the removal of 52 duplicate results across all three databases, a total of 601 articles were downloaded. Articles were sourced following Preferred Reporting Items for Systematic Reviews and Meta-Analyses (PRISMA) guidelines [32]. Figure 1 shows the flow diagram detailing the review process and study selection based on the PRISMA flow chart.

One major methodological concern throughout the literature related to long COVID treatment paradigms is the variability in post-infection time-period of the participants. Many interventions were provided to patients in the acute phase of the illness (e.g., fewer than 4 weeks post-infection) or to those fewer than 12 weeks post-infection. We utilized the World Health Organization’s definition of long COVID, characterized as symptoms persisting for more than 12 weeks after acute COVID-19 infection [2]. Recent guidance from the National Academies of Science, Engineering, and Medicine’s June 2024 consensus definition for long COVID concurred with the WHO definition [9]. This guidance includes five key elements in the definition of long COVID: (1) attribution to SARS-CoV-2 infection of any severity; (2) onset can be continuous from or delayed following acute SARS-CoV-2 infection, with the duration of symptoms present for more than 12 consecutive weeks; (3) symptoms are numerous and can range in severity and duration; (4) long COVID can affect anyone regardless of health, disability, or socioeconomic status, age, sex, gender, sexual orientation, race, ethnicity, or geographic location; and (5) long COVID can be resultant in profound emotional, cognitive, and physical impairments.

Consistent with the WHO’s and the National Academies of Science, Engineering, and Medicine’s June 2024 consensus definition, articles included in this review must have recruited adult participants who reported novel symptoms following SARS-CoV-2 infection that were consistently present for more than 12 weeks. The reported intervention must have been aimed at reducing symptomology associated with long COVID, and the authors must have discussed cognitive outcomes such as mental fatigue or brain fog, speed of processing, memory deficits, and/or executive functioning. After applying these stringent criteria, only 12 articles, including one case study and one case series, met these criteria (Table 1).

## 3. Review

### 3.1. Cognitive Training or Neuromodulation Interventions

Cognitive training with or without neuromodulation may assist in alleviating deficits in specific cognitive domains and are largely derivative from programs designed to remediate aging-related cognitive decline. Recently neurophysiologists have paired neuromodulatory techniques including transcranial magnetic, electrical, ultrasound, and optical stimulation as an adjuvant (or standalone) treatment with training. Five studies implemented a form of cognitive training or neuromodulation as the primary intervention strategy for participants struggling with the cognitive effects of long COVID. Four of these five studies utilized a paradigm for repetitive transcranial magnetic stimulation (rTMS).

Dunabeitia, et al. (2022) utilized a digital personalized computerized cognitive training (CCT) intervention to improve cognitive function among people living with long COVID. Participants completed the Cognitive Assessment Battery PRO (CAB), a self-administered online general cognitive evaluation psychometric tool developed by CogniFit Inc. (CogniFit Inc., San Franscisco, CA, USA) to assess and then later tailor participants’ 8-week CCT via the patented Individualized Training System™ (ITS) software that automatically chooses the activities and difficulty levels for each person in every session [33]. Specifically, performance scores were divided into five cognitive domains that comprise critical aspects of executive function: perception, attention, memory, coordination, and reasoning. Participants were asked to access the CAB-based training routinely for 8 weeks to improve their below-median baseline scores. A statistically significant effect of cognitive training was reported as participants who enrolled in CAB scored significantly higher in all five domains post-intervention. Importantly, coordination, operationalized via scores derived from response time and eye-hand coordination assessments, yielded the steepest increase in performance. Despite these promising results for a non-invasive, at-home intervention for people dealing with the cognitive fallout of long COVID, this study was only a feasibility study and included no control group, limiting the conclusions that can be drawn from these results due to the difficulty of isolating the true effect of the CCT and potential for bias from participant expectations.

Results from the remaining cognitive training or neuromodulation interventions utilized distinct rTMS protocols to target and remediate neurocognitive deficits associated with long COVID. It is important to note that two of these articles present data from case studies of one to four patients while the remaining two articles drew from a slightly larger pool of 14 to 23 patients (i.e., case series). Therefore, there were neither control groups nor effect size calculations included in these neuromodulatory interventions, limiting the conclusions that can be drawn and the ability to generalize findings to a wider population. The first of these case studies involved one patient, a 30-year-old Japanese woman who presented with chronic fatigue, cognitive dysfunction, and blurry vision that persisted for 7 months following SARS-CoV2 infection [37]. The authors did not report the specific rTMS paradigm that was used, but its results did align with those reported by Dunabeitia, et al., as their participant’s memory and reasoning functioning, measured via the Wechsler Adult Intelligence Scale (WAIS), specifically processing speed index score and perceptual reasoning index score, improved post-intervention (2022) [37]. Additionally, the subject’s verbal comprehension index and full-scale intelligence quotient increased from baseline, but her working memory index remained the same.

The second case series provided more methodological detail, with investigators utilizing three stimulation conditions: intermittent theta burst stimulation with 600 pulses (iTBS-600), iTBS-300, and sham stimulation [35]. Brain stimulation was delivered to the participants’ (*n* = 4) left dorsolateral prefrontal cortex in each of the three stimulation paradigms. It was reported that three of the four participants improved on a measure of behavioral inhibition (i.e., a modified Flanker interference assessing reasoning and processing speed) post-iTBS-600 protocol.

A study by Noda and colleagues reported a statistically significant increase in attention/concentration, retrospective and prospective memory, and planning/organizational abilities post-rTMS stimulation (2023). Unlike the participants of Sakib et al.’s case study, individuals of this pilot case series (*n* = 23) were asked to complete one session of iTBS for the left dorsolateral prefrontal cortex as well as one session of low frequency rTMS for the right lateral orbitofrontal cortex per day over the course of 20 days (for detailed protocol, see [45]). The addition of the second stimulation site was intended to better target the pathophysiology of long COVID. Cognitive function was assessed using the Perceived Deficits Questionnaire—Depression Five Item, generating the four sub scores used to assess the specific cognitive domains. Major limitations of this study were the lack of a control group, as this case series was an open-label preliminary trial for an rTMS intervention, and reliance on self-report versus objective cognitive outcome measures. These limitations increase the likelihood of a placebo effect and social desirability response by participants.

Sasaki and colleagues (2023), in another case series (*n* = 12), applied 1200 10-Hz rTMS to the midline of the occipital region followed by an application of 1200 10-Hz rTMS to the forehead (defined as 45° above the external auditory meatus) for a total of 10 applications to each neural area of interest. For Sasaki, et al.’s assessment of processing speed and overall cognitive function, the Wechsler Adult Intelligence Scale was implemented as in Tsuchida et al.’s rTMS case study (2023). Cognitive outcome measures included verbal comprehension, perceptual reasoning, working memory, and processing speed. A statistically significant increase in all four areas of cognitive function was reported post-intervention, along with an increased full scale intelligence quotient (FSIQ) from 94.6 ± 10.9 to 104.4 ± 13.0 [36].

These five studies reported an increase in processing speed and overall cognitive function, suggesting that neuropsychological deficits associated with long COVID can potentially be remediated. This finding speaks to the merit of incorporating a non-invasive, non-physically taxing (i.e., these paradigms require minimal physical exertion) component in an intervention. Patients with long COVID are particularly suited for such a rehabilitation strategy, as a high level of exercise intolerance is documented in this specific population [46,47]. Caution must be applied when interpreting these results, however, as none of the studies included a control group. Without a control group, it is difficult to confidently determine whether observed changes are due to the experimental intervention or other external factors, leading to unreliable conclusions about the effectiveness of the treatment being studied. Carefully controlled randomized control trials are warranted to support these promising interventions.

As to mechanism of action, the rTMS paradigm’s ability to increase cerebral blood flow may contribute to a recovery in cognitive function [48]. Both Tsuchida et al. and Sasaki et al. reported supporting findings from single-photon emission computed tomography (SPECT) in their participants, showing an overall increase in the level of blood flow within the brain after stimulation (2023). This change in vasculature has been documented in other published studies as well [48,49,50]. Another suggested mechanism for the positive impact of cognitive training and neuromodulation interventions is their ability to modulate synaptic connections in the brain [5,51,52,53]. As dynamic synapses are the basis for multiple cognitive processes, including learning and memory, the engagement and subsequent strengthening of synapses may help explain the reported improvement in cognitive functions. Changes in connectivity have been strongly linked to symptoms of brain fog and changes in overall connectivity as recently reviewed.

### 3.2. Exercise Interventions

A total of three studies that employed an exercise intervention and met the outlined methodological criteria were included in this narrative review. Each of the rehabilitation programs lasted for a different number of weeks and employed different exercise tactics. One of these interventions also included a psychological support component for participants, but, given the lack of a cognitive training aspect in the intervention, we have elected to treat this rehabilitation strategy as one that relied more heavily on the inclusion of physical activity. The uniting factor for these three studies is how they were designed with the patient’s diagnosis of long COVID in mind, helping to alleviate the documented exercise intolerance within this specific population [46,47].

The first of the three articles implemented a four-week physical rehabilitation strategy that included endurance training sessions (e.g., walking and ergocycling) and resistance training sessions [39]. Participants underwent a mean of 26 individualized endurance training sessions combining walking and ergocycling at an intensity close to the ventilation threshold, in addition to 12 individualized resistance training sessions. Patients were asked to work at a level of perceived exertion of 7 out of 10 and each resistance training session included four exercises: two exercises targeting the muscles of the lower limbs and two exercises targeting the muscles of the upper limbs. Originally intended for pulmonary rehabilitation, the training was adapted for effect on cognitive function in long COVID as measured by the Montreal Cognitive Assessment (MoCA), the Modified Fatigue Impact Scale (MFIS), and other related, validated questionnaires. Post-intervention, participants reported statistically significant lower levels of mental fatigue and scored significantly higher on the MoCA, demonstrating the promise of an exercise intervention for individuals with long COVID.

The second exercise intervention also used the MFIS as the outcome measure [40]. This intervention utilized a high-intensity interval training program that incorporated aerobic exercise with a warm-up and cool-down phase three times a week across six weeks. There was no control group included in this study, making it difficult to isolate the impact of the intervention from other potential influences; however, there was a statistically significant decline in both male and female participants’ MFIS scores post-exercise intervention, demonstrating a decrease in overall levels of fatigue and in the overall impact of fatigue.

The third and final article was conducted by McGregor, et al., (2024). As a part of the Rehabilitation Exercise and psycholoGical support After COVID-19 InfectioN (REGAIN) trial, 585 adults (26–86 years) with ongoing physical and/or mental health sequelae secondary to long COVID were included in the multicenter, parallel-group, randomized controlled trial. Participants were randomized to receive the REGAIN intervention (exercise and psychological support; *n* = 298) or usual care (*n* = 287). The REGAIN intervention consisted of an eight-week course of online, weekly, home-based, live, supervised group exercise and psychological support sessions. Usual care was a single online educational session. The Patient-Reported Outcomes Measurement Information System (PROMIS) health-related quality of life was the primary outcome measure. Secondary outcomes included PROMIS subscores of depression, fatigue, sleep disturbance, pain interference, physical function, social roles/activities, and cognitive function. The REGAIN participants reported a statistically significant improvement in health-related quality of life at 3 months, which was sustained at 12 months compared to usual care, along with statistically significant improvements in self-reported depression, fatigue, and sleep disturbance. By 12 months, all PROMIS subscores and subscales were improved more in the intervention group; however, the improvements were not statistically significant in the domains of pain interference physical function, social roles/activities, or cognitive function. These findings do highlight the need for rigorous studies aimed specifically at cognitive rehabilitation for patients struggling with long COVID.

### 3.3. Pharmacological Interventions

Four articles met this narrative review’s criteria and implemented a form of a pharmacological intervention, including antihistamine drugs, a novel neuroprotectant drug complex, and aromatherapy. Salvucci and colleagues chose to administer two antihistamine medications to the 14 participants in their experimental group: 180 mg of fexofenadine (histamine H1 receptor antagonist) and 40 mg of famotidine (histamine H2 antagonist) each day for 20 days (2023). The 13 participants in the control group were not treated with these antihistamines as they refused further pharmacological treatment for their symptoms. The decision to administer fexofenadine and famotidine was built upon the hypothesis that hyperinflammation, mediated partly by the activation of mast cells, contributes to the persistence of COVID-like symptoms beyond the initial infection period [54,55,56,57]. It was hypothesized that blocking the H1 and H2 receptors with these commonly prescribed medications would relieve long COVID-associated symptoms such as brain fog and fatigue. Importantly, participants in this study self-reported their level of mental clouding or brain fog, generating the major limitation of the study, as no verified questionnaire or cognitive assessment was administered. Regardless, a significant portion of the participants in the experimental group reported a vanishing of fatigue (43%) and brain fog (43%) post-intervention. This is in comparison to those in the control group, who reported no statistically significant improvements across symptoms.

The second pharmacological intervention included here investigated the effects of Promomed’s new drug Brainmax [44]. Brainmax is comprised of the coordination complex between ethylmethylhydroxypyridine and trimethylhydrosinium propionate with succinate acid anion and was shown to possess a good safety profile in a 2022 randomized, double-blind, placebo-controlled study [44]. Fifteen participants received an intramuscular injection of Brainmax for 10 days while 15 participants received a placebo injection. Cognitive function was assessed using the Multidimensional Fatigue Inventory (MFI-20) and the Montreal Cognitive Assessment (MoCA). A statistically significant reduction in MFI-20 scores and a slight increase in MoCA scores were reported for patients who received Brainmax injections when compared to the placebo group. The authors suggested that a potential mechanism underlying the “neuroprotective, neuroregenerative, and neuroactivating” influence of Brainmax could be the complex’s influence on the mitochondria as it enhances the stability of mitochondrial function [44,58]. Specifically, it reduces the severity of oxidative stress and the intensity of reactive oxidative species formation [59].

Additionally, the authors collected structural and functional magnetic resonance images (fMRI) before and after the pharmacological treatment and reported an increase in functional connectivity between the left dorsolateral prefrontal cortex and the proximal part of the left temporal operculum. Therefore, the decrease in physical and mental fatigue as well as the increase in MoCA score could be partially explained by the increase in correlations between neural areas critical for the processing and integration of incoming stimuli seen in individuals treated with Brainmax.

The third and final traditional pharmaceutical intervention utilized a palmitoylethanolamide-luteolin combination drug (PEA-LUT) [41]. Like the two drugs employed by Salvucci and colleagues, PEA-LUT has well-documented anti-inflammatory properties largely due to its ability to inhibit the release of mast cells and counteract the negative effects of cytokines, making it a strong candidate drug for relieving cognitive symptoms associated with long COVID [60,61]. De Luca and colleagues administered this supplement consistently for 90 days alone and in conjunction with olfactory training to individuals diagnosed with long COVID. No control group (i.e., no placebo was administered to participants with or without olfactory training) was included in this longitudinal study. Participants’ levels of brain fog were measured by a previously published questionnaire comprising four detailed questions that assessed a person’s concentration and overall mental ability [62].

Post-treatment, a non-statistically significant reduction in brain fog was observed for participants who received PEA-LUT alone. Participants who received PEA-LUT and olfactory training achieved a statistically significant reduction in brain fog. While an explanation was not offered as to why statistical significance was only achieved with the combination approach (i.e., PEA-LUT with olfactory training), the authors did cite the ability of palmitoylethanolamide to modulate histamine release and the ability of luteolin to promote the antioxidant response in neurons for the supplement’s influence over the prevalence of brain fog [41].

A final experimental protocol that chose a non-traditional pharmaceutical approach investigated the efficacy of an alternative medicinal treatment in a randomized, double-blinded, placebo controlled clinical trial: aromatherapy [42]. The authors of this paper hoped to relieve a multitude of symptoms associated with long COVID, but their main symptom of interest was mental fatigue. Outcomes were assessed using the multidimensional fatigue symptom inventory—short form (MFIS-SF). The essential oils included for the intervention group included thyme (*Thymus vulgaris*), orange peel (*Citrus sinensis*), clove bud (*Eugenia caryophyllus*), and frankincense (*Boswellia carterii*) and were produced by Young Living Essential Oils. After a two-week period of exposure to the prescribed blend of essential oils, a statistically significant reduction in mental fatigue was reported, highlighting how non-traditional medicinal interventions can be used alongside their traditional pharmacological intervention counterparts.

## 4. Conclusions

Of particular interest to this narrative review was treatment remediation of long COVID-associated neurocognitive deficits, such as brain fog, persistent mental fatigue, and speed of processing, that are estimated to affect as many as 40% of patients currently struggling with long COVID [6,31].

The 12 articles collected for this narrative review present a diverse collection of rehabilitation strategies aimed at ameliorating long COVID-associated neurocognitive deficits. Treatment interventions such as cognitive training, rTMS, exercise rehabilitation, and pharmacological and other related treatment paradigms were found to be effective in reducing long COVID-related cognitive issues. While many of these studies have significant limitations (e.g., lack of an appropriate control group, use of self-report versus objective cognitive measures, unclear methodology), all reported a reduction in neurocognitive symptomology of long COVID to varying degrees post-intervention (i.e., some interventions resulted in statistically significant improvements, while others did not). Across the collected articles, reductions in mental fatigue, operationalized via self-reported measures and various assessments, and improvements in processing speed and brain fog were reported. Other promising therapeutic interventions for post-COVID neurocognitive deficits and fatigue include low dose naltrexone (LND) and hyperbaric oxygen therapy (H-BOT) [63,64,65]. Additionally, Uswatte et al. (2024) describe a pilot RCT (*n* = 14) using Constraint-Induced Cognitive Therapy (CICT) for the treatment of cognitive difficulties for individuals with long COVID [66]. The intervention resulted in large, statistically significant improvements in brain fog symptoms and performance of everyday activities. The investigators also found that 80% of those who were unemployed before CICT returned to work after the experimental intervention, while no one from the treatment-as-usual control group returned to work.

Planned randomized controlled studies using these therapeutic interventions with long COVID patients meeting the WHO and the National Academies of Science, Engineering, and Medicine’s June 2024 consensus definition of long COVID will be of interest. The material was presented in a narrative review format to provide a comprehensive overview of currently published long COVID interventions. The findings of this review indicate promise for brief, non-invasive interventions for long COVID brain fog and mental fatigue and serve as a basis to inform a cohesive strategy for future treatment intervention studies.

The dearth of well-controlled studies highlights the need for continued investigation of effective interventions aimed at reducing the cognitive deficits associated with long COVID. Overall, the findings from this narrative review indicate that there are very few tightly controlled experimental designs and thus few definitive conclusions that can be drawn from these reports. Several research limitations were inherent to many of the studies in this review, including absence of control group, low sample sizes, no reported effect size calculations, outcome measures with low sensitivity, specificity, reliability, and validity, and limited reporting on participant variables and details of the interventions. Long COVID is persistent, debilitating, and enduring. Patients and practitioners are looking for safe interventions and a call to action is necessitated, as in the absence of safe, research-driven interventions patients may seek alternative and possibly harmful health alternatives.

Fatigue, manifesting as both physical fatigue and mental fatigue, is a core feature of long COVID. Fatigue is multifactorial and multisymptomatic and can be a driver of other physical, cognitive, and emotional issues. Future studies that employ a multi-modal approach to address fatigue, cognition, and mental health issues with appropriate controls are needed, along with descriptions of adjuvant, appropriate medical treatments targeted at specific symptom clusters.

Additionally, future research should direct more attention to the post-infection time frame of ≥12 weeks post last COVID-19 infection, consistent with the National Academies of Science, Engineering, and Medicine’s consensus definition for long COVID [9]. Several articles were excluded from this narrative review as the authors failed to report an infection timeline for their participants, thus reducing the implied efficacy of the intervention versus spontaneous recovery in the acute post-infection phase. By adhering to the standard ≥12 weeks post-infection length of time as an inclusion criterion, the label of long COVID could be applied in a more controlled manner and ensure that patient populations are comparable in terms of their disease progression.

In conclusion, long COVID (regardless of infection severity or number of infections) is a global public health challenge with a sizeable effect on patient quality of life, economic productivity, and ultimately the global economy. Interventions aimed at improving fatigue, processing speed, and other related cognitive outcomes are urgently needed. Given the lack of experimental studies and randomized controlled trials (RCTs) focused on long COVID interventions, this review can serve as a call to action for clinicians and researchers.

## Figures and Tables

**Figure 1 biomedicines-13-00421-f001:**
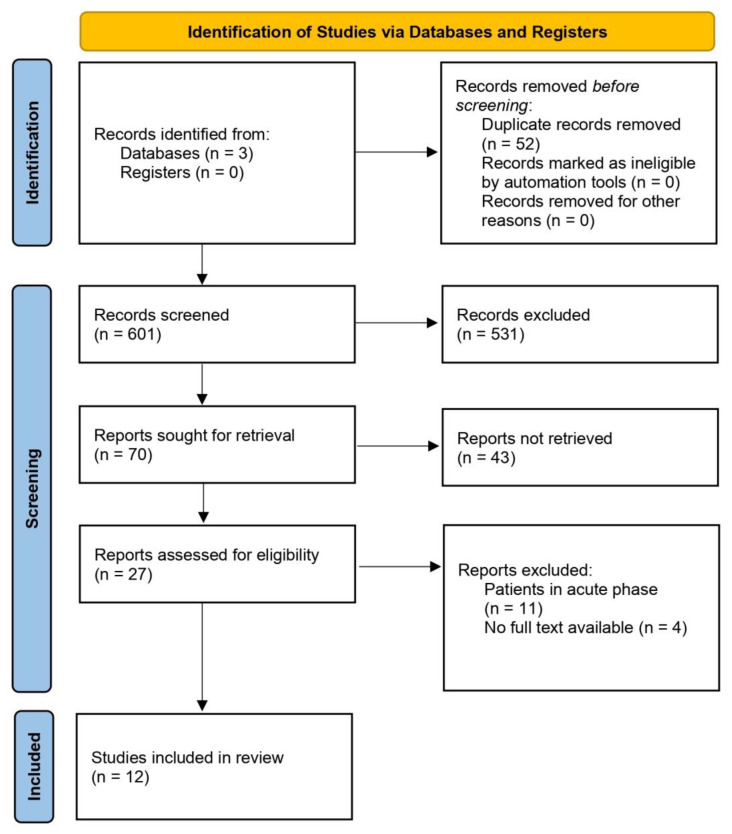
PRISMA flow chart.

**Table 1 biomedicines-13-00421-t001:** Articles included in this narrative review.

Author and Year	Number of Participants	Type of Intervention	Study Design	Reported Results
Dunabeitia et al., 2022 [33]	73	Cognitive Training	Feasibility Study	Sig. increase in perception, attention, memory, coordination, and reasoning skills
Noda et al., 2023 [34]	23	Cognitive Training (rTMS)	Pilot Study	Sig. increase in attention, memory, and planning/organizational skills
Sakib et al., 2024 [35]	4	Cognitive Training (rTMS)	Case Series	Non-sig. increase in behavioral inhibition performance
Sasaki et al., 2023 [36]	12	Cognitive Training (rTMS)	Interventional Study	Sig. increase in verbal comprehension, reasoning, working memory, and processing speed performance
Tsuchida et al., 2023 [37]	1	Cognitive Training (rTMS)	Case Study	Non-sig. increase in memory and reasoning skills
McGregor et al., 2024 [38]	585	Exercise	Randomized Controlled Trial	Non-sig. improvement in cognitive function
Moine et al., 2024 [39]	47	Exercise	Interventional Study	Sig. increase in MoCA scores and sig. reduction in mental fatigue
Rzepka-Cholasinska et al., 2024 [40]	90	Exercise	Observational Study	Sig. reduction in mental fatigue
De Luca et al., 2022 [41]	69	Pharmacological (PEA-LUT)	Longitudinal Interventional Study	Non-sig. reduction in mental clouding with drug treatment alone
Hawkins et al., 2022 [42]	40	Pharmacological (Aromatherapy)	Randomized Controlled Trial	Sig. reduction in mental fatigue
Salvucci et al., 2023 [43]	27	Pharmacological (Antihistamines)	Interventional Study	Sig. reduction in self-reported brain fog
Tanashyan et al., 2023 [44]	30	Pharmacological (Brainmax)	Randomized Controlled Trial	Sig. reduction in mental fatigue

## Data Availability

No new data were created or analyzed in this study.

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
