# Peer review of "A Narrative Review of the Efficacy of Long COVID Interventions on Brain Fog, Processing Speed, and Other Related Cognitive Outcomes"

_biomedicines, 2025, doi:10.3390/biomedicines13020421_

Round 1
Reviewer 1 Report
Comments and Suggestions for Authors
As you said in the abstract, the article is a "narrative review", so please mention it in the title.
Indicate in the abstract the time period in which the articles were searched and the date of the articles reviewed.
What type of studies exactly did you review? Please mention the type of studies in the abstract.
At the end of the abstract; the general conclusion and main finding of the article should be mentioned.
The introduction is well written and can provide the reader with good information.
In Table 1; it is better to mention more information from the articles. For example; type and method of the study; duration of the intervention; age of the participants and... . It seems that the article continues to talk well about the research variables and one of the most important things that needs to be corrected; is to mention much more details in this table.
Research limitations should be mentioned at the end of the article. For example; lack of experimental studies and RCTs
Author Response
Thank you very much for taking the time to review this manuscript. Please find detailed responses below and the corresponding revisions/corrections highlighted/in track changes in the re-submitted files. We feel that we have addressed the comments carefully and to the best of our ability and that the result is a much-improved manuscript.
REVIEWER 1: Comments and Suggestions for Authors
|
Comments 1: As you said in the abstract, the article is a "narrative review", so please mention it in the title. |
|
Response 1: Thank you for pointing this out. We agree with this comment. Therefore, we have changed the title to “A Narrative Review of the Efficacy of Long COVID Interventions on Brain Fog, Processing Speed, and Other Related Cognitive Outcomes”. This is reflected on page 1, Line 2 of the revised manuscript. Comments 2: Indicate in the abstract the time period in which the articles were searched and the date of the articles reviewed. Response 2: Yes – we agree this is important information to include in the abstract. The search for this narrative review was completed in June of 2024, and articles between the dates of January 2020 (onset of the pandemic) and June 2024 were included in the narrative review, with the first article published in 2022. We accordingly have included the following sentence in the abstract of the manuscript on page 1, Lines 28 and 29: Articles published between the dates of 01/2020 and 6/30/2024 were included in the search. This sentence was repeated in the Materials and Methods section on page 3, Lines 135 and 136, for clarity. Given the length of time that has passed between the search and the submission of this manuscript, an updated review of the literature at the time of this revision (January 23, 2025), revealed no new additions to the literature that would meet the stringent criteria for this narrative review. For inclusive findings, the following was added to the Conclusion section of the manuscript, page 9 and 10, Lines 416-428 and additional references added to the References section on pages 12-14. Other promising therapeutic interventions for post-COVID neurocognitive deficits and fatigue include low dose naltrexone (LND) and hyperbaric oxygen therapy (H-BOT); (Naik et. al, 2024; Tamariz et. al, 2024; Zilberman-Itskovich et. al, 2022). Additionally, Uswatte et. al (2024) describe a pilot RCT (N=14) using Constraint-Induced Cognitive Therapy (CICT) for the treatment of cognitive difficulties for individuals with long COVID. The intervention resulted in large, statistically significant improvements in brain fog symptoms and performance of everyday activities. The investigators also found that 80% of those who were unemployed before CICT returned to work after the experimental intervention, while no one from the treatment-as-usual control group re-turned to work. Planned randomized controlled studies using these therapeutic interventions with long-COVID patients meeting the WHO and National Academies of Science, Engineering, and Medicine’s June 2024 consensus definition of long COVID will be of interest.
|
Comments 3: What type of studies exactly did you review? Please mention the type of studies in the abstract.
Response 3: We agree this is also important information to include in the abstract. This addition was added to the sentence on page 1, Lines 29-32 in the Abstract: . Twelve studies were included in the narrative review, including a feasibility study, a pilot study, a case series, a case study, an observational study, in addition to three randomized clinical trials and four interventional studies. Additionally, this sentence was added on page 3, Lines 136-139: The aspirational goal was to include only randomized controlled clinical trials in this review; however, given the dearth of literature examining interventions for long COVID cognitive issues, a variety of methodologies were considered for inclusion.
Comments 4: At the end of the abstract; the general conclusion and main finding of the article should be mentioned.
Response 4: The authors are in agreement and changed the final sentences in the Abstract on page 1, Lines 32-37 to: Overall, treatment interventions such as cognitive training, noninvasive brain stimulation therapy, exercise rehabilitation, pharmacological, and other related treatment paradigms show promise in reducing long COVID cognitive issues. This narrative review highlights the need for more rigorous experimental designs and future studies are needed to fully evaluate treatment interventions for persistent cognitive deficits associated with long COVID.
Comments 5: The introduction is well written and can provide the reader with good information.
Response 5: Thank you.
Comments 6: In Table 1; it is better to mention more information from the articles. For example; type and method of the study; duration of the intervention; age of the participants and... . It seems that the article continues to talk well about the research variables and one of the most important things that needs to be corrected; is to mention much more details in this table.
Response 6: We agree with the suggestion that more detail should be included in Table 1. In response to this suggestion, an additional column was added to Table 1 to include “Study Design”. The authors also agree that adding columns to reflect duration of the intervention, age of participants, etc. would have been beneficial; however, after further consideration it was discovered that specific participant demographics and intervention length were rarely included in the articles included in this review. That said, we do believe Table 1 now fulfills its designation as a summary table. Please see the updated Table 1 in the Supplementary Files Section and notation to input the updated table on page 4, Line 169.
Comments 7: Research limitations should be mentioned at the end of the article. For example; lack of experimental studies and RCTs
Response 7: We are in agreement with this suggestion and the following sentence was added to page 10, Lines 437-440: Several research limitations were inherent to many of the studies in this review, including absence of control group, low sample sizes, no reported effect size calculations, outcome measures with low sensitivity, specificity, reliability, and validity, and limited reporting on participant variables and details of the interventions. With an additional sentence added to the last line of the manuscript on page 10, Lines 463-466. Given the lack of experimental studies and randomized controlled trials (RCTs) focused on long COVID interventions, this review can serve as a call to action for clinicians and researchers.
Reviewer 2 Report
Comments and Suggestions for Authors
Whitaker-Hardin et al. conducted a narrative review of long COVID interventions on some cognitive outcomes, such as brain fog. I have some major concerns regarding the study design of this manuscript:
- The paper summarizes the findings rather than critically evaluating them. For example, the small sample sizes and lack of placebo controls are noted but not discussed in depth or tied to how they affect the reliability of the conclusions.
- Throughout the manuscript, terms like “significant increase” are used. Can you clarify whether outcomes were statistically significant and describe the statistical tests used in the original studies? If they are merely observed changes, I would doubt their validity.
- The paper selection is not stringent. Some papers included are preprints (Dunabeitia et al., 2022), while others have very few subjects, such as Tscuchida et al., 2023 (N =1) or Sakib et al., 2024 (N =4).
- Were any meta-analysis approaches considered to combine results?
- A PRISMA flow diagram is missing.
- The supplementary 1 cannot be opened.
Author Response
Thank you very much for taking the time to review this manuscript. Please find detailed responses below and the corresponding revisions/corrections highlighted/in track changes in the re-submitted files. We feel that we have addressed the comments carefully and to the best of our ability and that the result is a much-improved manuscript.
REVIEWER 2: Comments and Suggestions for Authors
Whitaker-Hardin et al. conducted a narrative review of long COVID interventions on some cognitive outcomes, such as brain fog. I have some major concerns regarding the study design of this manuscript:
Comments 1/Reviewer 2: The paper summarizes the findings rather than critically evaluating them. For example, the small sample sizes and lack of placebo controls are noted but not discussed in depth or tied to how they affect the reliability of the conclusions.
Response 1/Reviewer 2: We agree that a more critical review improves the overall quality of the paper. As such, we have added several sentences to address this issue.
We added the following statement on page 5, Line 198-199: due to the difficulty of isolating the true effect of the CCT and potential for bias from participant expectations.
We made the following addition page 5, 206-207: and limited ability to generalize findings to a wider population.
This sentence was edited and we added a second sentence on page 6, Lines 233-237: Major limitations of this study were the lack of control group as this case series was an open-label preliminary trial for an rTMS intervention and reliance on self-report versus objective cognitive outcome measures. These limitations increase the likelihood of a placebo effect and social desirability response by participants.
This sentence was added on page 6, Lines 255-258: Without a control group, it is difficult to confidently determine whether observed changes are due to the experimental intervention or other external factors, leading to unreliable conclusions about the effectiveness of the treatment being studied.
This addition was added on page 7, Lines 301-304: There was no control group included in this study, making it difficult to isolate the impact of the intervention from other potential influences, however
Comments 2/Reviewer 2: Throughout the manuscript, terms like “significant increase” are used. Can you clarify whether outcomes were statistically significant and describe the statistical tests used in the original studies? If they are merely observed changes, I would doubt their validity.
Response 2/Reviewer 2: We have added the appropriate qualifiers throughout the manuscript, as follows:
statistically significant, page 5. Line 191
statistically significant, page 6, Line 224.
A statistically significant, page 6, Lines 244-245.
there was a statistically significant decline in both male and female participants’ MFIS scores post-exercise intervention, page 7, Lines 303-304.
statistically significant, page 7, Line 317.
statistically significant, page 8, Line 345.
statistically significant, page 8, Line 353-354.
statistically significant, page 9, Line 398.
Comments 3/Reviewer 2: The paper selection is not stringent. Some papers included are preprints (Dunabeitia et al., 2022), while others have very few subjects, such as Tscuchida et al., 2023 (N =1) or Sakib et al., 2024 (N =4).
Response 3/Reviewer 2: The authors are in agreement with this criticism, and we were surprised at the dearth of studies available in the literature examining interventions for long COVID cognitive issues. To further highlight this issue, the following sentence was added on page 3, Lines 136-139: The aspirational goal was to include only randomized controlled clinical trials in this review; however, given the dearth of literature examining interventions for long COVID cognitive issues, a variety of methodologies were considered for inclusion. Additionally, the following sentence was added to page 10, Lines 437-440: Several research limitations were inherent to many of the studies in this review, including absence of control group, low sample sizes, no reported effect size calculations, outcome measures with low sensitivity, specificity, reliability, and validity, and limited reporting on participant variables and details of the interventions. With an additional sentence added to the last line of the manuscript on page 10, Lines 463-466. Given the lack of experimental studies and randomized controlled trials (RCTs) focused on long COVID interventions, this review can serve as a call to action for clinicians and researchers.
Comments 4/Reviewer 2: Were any meta-analysis approaches considered to combine results?
Response 4/Reviewer 2: Yes, meta-analysis approaches were considered; however, unfortunately, few of the studies provided quantitative outcomes. The following sentence was added to the manuscript on page 3, Lines 117-122 to clarify: The authors chose to conduct a narrative review, as few papers that focused on the treatment of cognitive issues in long COVID included comparable outcome measures or detailed quantitative outcomes, making it impossible to extract and standardize data from each study to statistically pool the data to generate an estimate of the effect size across the studies.
Comments 5/Reviewer 2: A PRISMA flow diagram is missing.
Response 5/Reviewer 2: Thank you for identifying this oversight. We have added a PRISMA flow diagram in response to this. Please see the Figures Graphics Images section for this diagram. We also added the following sentences to page 3, Lines 140-143, along with the citation in the reference section: Articles were sourced following Preferred Reporting Items for Systematic Reviews and Meta-Analyses (PRISMA) guidelines (Page et al., 2021). Figure 1 shows the flow diagram detailing the review process and study selection based on the PRISMA flow chart.
Comments 6/Reviewer 2: The supplementary 1 cannot be opened.
Response 6/Reviewer 2: Our apologies. We are hopeful this issue has been addressed by the journal editors.
Round 2
Reviewer 2 Report
Comments and Suggestions for Authors
I appreciated the authors for addressing all my concerns. I have no further comments. Thanks.